# Effect of Anti-Rheumatic Treatment on the Periodontal Condition of Rheumatoid Arthritis Patients

**DOI:** 10.3390/ijerph18052529

**Published:** 2021-03-04

**Authors:** Menke J. de Smit, Johanna Westra, Marcel D. Posthumus, Gerald Springer, Arie Jan van Winkelhoff, Arjan Vissink, Elisabeth Brouwer, Marc Bijl

**Affiliations:** 1Department of Oral and Maxillofacial Surgery, University of Groningen and University Medical Center Groningen, P.O. Box 30.001, 9700 RB Groningen, The Netherlands; m.j.de.smit@umcg.nl; 2Department of Rheumatology and Clinical Immunology, University of Groningen and University Medical Center Groningen, P.O. Box 30.001, 9700 RB Groningen, The Netherlands; johanna.westra@umcg.nl (J.W.); e.brouwer@umcg.nl (E.B.); 3Center for Dentistry and Oral Hygiene, University of Groningen and University Medical Center Groningen, Antonius Deusinglaan 1, 9713 AV Groningen, The Netherlands; ajwinkelhoff@laboral.nl; 4Department of Internal Medicine and Rheumatology, Martini Hospital Groningen, Van Swietenplein 1, 9728 NT Groningen, The Netherlands; m.d.posthumus@mzh.nl (M.D.P.); m.bijl@mzh.nl (M.B.); 5Department of Oral and Maxillofacial Surgery, Martini Hospital Groningen, Van Swietenplein 1, 9728 NT Groningen, The Netherlands; g.springer@mzh.nl

**Keywords:** rheumatoid arthritis, periodontitis, treatment, anti-TNF, methotrexate

## Abstract

Periodontitis, a bacterial-induced infection of the supporting soft and hard tissues of the teeth (the periodontium), is common in patients with rheumatoid arthritis (RA). As RA and periodontitis underlie common inflammatory pathways, targeting the progression of RA might mediate both periodontitis and RA. On the other hand, patients with RA on immunosuppressive medication have an increased risk of infection. Therefore, the objective of this longitudinal observation study was to assess the effect of methotrexate (MTX) and anti-tumor necrosis factor-α (anti-TNF, etanercept) treatment on the periodontal condition of RA patients. Overall, 14 dentate treatment-naive RA patients starting with MTX and 12 dentate RA patients starting with anti-TNF therapy in addition to MTX were included. Follow-up was scheduled matching the routine protocol for the respective treatments. Prior to the anti-rheumatic treatment with MTX or the anti-TNF therapy in addition to MTX, and during follow-up, i.e., 2 months for MTX, and 3 and 6 months for the anti-TNF therapy in addition to MTX, the periodontal inflamed surface area (PISA) was measured. The efficacy of the anti-rheumatic treatment was assessed by determining the change in RA disease activity (DAS28-ESR). Furthermore, the erythrocyte sedimentation rates were determined and the levels of C-reactive protein, IgM-rheumatoid factor, anti-cyclic citrullinated protein antibodies, and antibodies to the periodontal pathogen *Porphyromonas gingivalis,* were measured. Subgingival sampling and microbiological characterization of the subgingival microflora was done at baseline. MTX or anti-TNF treatment did not result in an improvement of the periodontal condition, while both treatments significantly improved DAS28 scores (both *p* < 0.01), and reduced C-reactive protein levels and erythrocyte sedimentation rates (both *p* < 0.05). It is concluded that anti-rheumatic treatment (MTX and anti-TNF) has negligible influence on the periodontal condition of RA patients.

## 1. Introduction

Periodontal disease is associated with a chronic inflammatory response due to a dysbiotic subgingival microbial ecology resulting in the destruction of the teeth-supporting soft and hard tissues (periodontium) [1]. Aside from local tissue destruction, continuous low-grade inflammation may have systemic consequences. In this respect, a link between existence and/or severity of periodontal disease and systemic diseases, such as diabetes, cardiovascular disease, and rheumatoid arthritis (RA), has been shown [2,3].

RA is a systemic autoimmune disease characterized by chronic inflammation of the synovial joints (synovitis), leading to the irreversible destruction of cartilage and bone tissues. Clinical signs in RA patients include pain, swelling, and stiffness of the joints. RA occurs in about 0.5–1% of the world population and is therefore the most common autoimmune disease in the world [4]. About two-thirds of RA patients suffer from moderate or severe periodontitis [5,6].

Overlapping molecular pathways of inflammation underlie periodontitis and RA [7]. In both diseases, local tissue destruction involves the production of inflammatory cytokines and proteolytic proteins, such as matrix metalloproteinases. The presence of shared underlying inflammatory pathogenesis mediating the progression of both periodontitis and RA could provide potentially common therapeutic targets [8]. Traditional approaches to control RA rely on conventional synthetic disease-modifying anti-rheumatic drugs (csDMARDS), such as methotrexate (MTX). MTX is a folate analogue with inhibitory effects on interleukin-1 (IL-1) and tumor necrosis factor-α (TNF) production. Advances in understanding key events in the pathogenesis of RA have led to development of biological DMARDs [4]. The most validated and effective therapy blocks or antagonizes the actions of TNF, a cytokine which is considered to orchestrate the inflammatory response in RA [9]. Apart from its well-documented therapeutic value, TNF inhibition is associated with an increased risk of infectious adverse events [10].

Currently, while csDMARDS and biological DMARDS have shown their efficacy in the treatment of RA, insufficient evidence is available to estimate the effect of such anti-rheumatic treatment on the periodontal condition of RA patients [11,12]. There is some evidence that treatment with anti-TNF might initially improve bleeding on probing and the gingival index, and later on it might improve bleeding on probing pocket depth and clinical attachment level [13]. On the contrary, cross-sectional studies revealed that anti-rheumatic treatment with csDMARDS and biological DMARDS is associated with periodontal inflammation [14,15], with the combination of MTX and anti-TNF showing an increased potential for periodontal inflammation [14]. Therefore, the purpose of this study was to determine the effect of MTX and anti-TNF (etanercept) on the periodontal condition, as assessed by the periodontal inflamed surface area (PISA) [16]. Since good oral health is important for a patient’s well-being, information on the effects of treatment are important.

## 2. Materials and Methods

### 2.1. Patients

Consecutive patients visiting the outpatient clinic of the Department of Internal Medicine and Rheumatology of the Martini Hospital Groningen, who were eligible for starting MTX treatment (newly diagnosed) or MTX combined with anti-TNF (etanercept) treatment, were included. Inclusion criteria were being >18 years of age, fulfilling the American College of Rheumatology/European League Against Rheumatism ACR/EULAR criteria for RA (2010) [17], and a willingness to participate and written informed consent. Exclusion criteria were edentulism, infection other than periodontitis, inflammation other than RA, present malignancy, diabetes, active thyroid disease, myocardial infarction, stroke or recanalization of the femoral arteries for claudication less than 6 months prior to the study, pregnancy including a 6-months postpartum period, breastfeeding, malnutrition, alcoholism, drug abuse, use of corticosteroids >10 mg/day, and antibiotic use less than 3 months prior to the study. Follow-up was according to the routine follow-up scheme for patients on anti-rheumatic treatment (2 months for MTX, 3 and 6 months for anti-TNF). During the study, no dental or periodontal treatment was allowed.

During this study, no periodontal treatment was allowed, as the primary goal of periodontal treatment is to control microbial periodontal infection by removing the (subgingival) bacterial biofilm by mechanical debridement and creating circumstances for optimal bacterial plaque control by the patient. Such a treatment would interfere with the outcome of the current study, as active periodontal treatment includes scaling and root planing with ultrasonic or hand instruments, and can also include surgical therapy, extraction of hopeless teeth, and/or use of local or systemic antiseptic or antibiotic measures. Periodontal therapy triggers a short-term inflammatory response followed by a progressive and consistent reduction of systemic inflammation [18].

The study protocol was approved and released from Medical Research Involving Human Subjects Act duty by the Medical Ethical Committee (METc 2013/112).

### 2.2. Clinical Parameters

At baseline, i.e., before the MTX or MTX combined with anti-TNF treatment, and during the routine follow-up, RA disease activity (DAS28-ESR) was scored by two well-trained research nurses. At the same time, the periodontal condition was evaluated by a periodontist blinded for RA treatment. The periodontal condition was assessed using full-mouth oral measurements (probing pocket depth, bleeding on probing), and the periodontal inflamed surface area (PISA) [16] was calculated. The PISA reflects the extent of inflammatory burden due to periodontal inflammation and is, therefore, considered an adequate measure to assess the impact of anti-rheumatic treatment on the periodontal condition. A PISA value ≥1.30 cm^2^ has been shown to be associated with the Centers for Disease Control and Prevention—American Academy of Periodontology (CDC-AAP) case definition of periodontitis [19]. Full-mouth plaque scores (the percentage of sites with visible plaque, scored at six sites per tooth) were measured to estimate the level of oral hygiene. At baseline, a panoramic radiograph was taken to confirm clinical findings (alveolar bone loss, furcation involvement, caries) and to diagnose peri-apical pathology.

### 2.3. Sampling

Peripheral blood was drawn at baseline and at every follow-up visit. Subgingival sampling and microbiological characterization of the subgingival microflora was done at baseline, i.e., at most one week before the start of therapy, using standard sampling procedures and anaerobic culture techniques (determination of the presence and proportions of the established periodontal pathogens *Porphyromonas gingivalis (Pg), Prevotella intermedia*, *Fusobacterium nucleatum*, *Parvimonas micra*, *Tannerella forsythia*, *Campylobacter rectus*, and other dark-pigmented gram-negative anaerobic rods). Briefly, subgingival samples were taken by using sterile paper points. Microbiological sampling included the selection of the deepest bleeding periodontal pocket in each quadrant of the dentition on the basis of pocket probing depth measurements. If there were no bleeding periodontal pockets, the mesial sites of the first molars, or, in the absence of a first molar, the mesial sites of the adjacent anterior tooth in the dental arch, were selected. The sample sites were isolated with cotton rolls, and supragingival plaque was carefully removed with curettes and cotton pallets. Subsequently, two paper points were inserted to the depth of the pocket and left in place for 10 s. All paper points per subject were pooled in reduced transport fluid and processed for microbiological examination immediately after sampling [5]. 

### 2.4. Laboratory Procedures

The erythrocyte sedimentation rate (ESR) was determined routinely. Serum was investigated for C-reactive protein (CRP) concentration by enzyme-linked immunosorbent assay (ELISA) (DuoSet; R&D Systems, Minneapolis, MN, USA). The IgM-rheumatoid factor (IgM-RF), in international units per milliliter) was measured by using an in-house validated ELISA (cut-off point for positivity: 25 IU/mL) [20]. The IgG anti-cyclic citrullinated protein antibody (anti-CCP) levels were measured using a commercial anti-CCP2 kit (Euro Diagnostica, Malmö, Sweden) according to the manufacturer’s protocol (cut-off point for positivity: 25 U/mL). As *Pg* has been proposed to play a crucial role in the initiation or propagation of periodontitis-associated RA [21], subgingival prevalence as well as antibodies against this bacterium were considered. IgG antibody levels against *Porphyromonas gingivalis* (anti-*Pg*) were determined by in-house ELISA using a pooled lysate of clinical isolates of *Pg* as the antigen and IgG standard curves, as described previously [5]. The absorbance was read at 450 nm in a Versamax Microplate Reader, and the antibody levels were calculated using SoftMax^®^ Pro 5 software (Molecular Devices, San Jose, CA, USA).

### 2.5. Statistical Analysis

As it was not known how large the potential effect of treating RA patients with csDMARDS or biological DMARDS might be on the inflammatory burden because of periodontal inflammation, no formal power analysis was performed at the start of the study. However, it was decided to assess whether such a treatment might result in a clinically relevant improvement of the periodontal condition in a convenient sample of RA patients when applying the current best method (PISA) for assessing the impact of the inflammatory burden due to periodontal inflammation. 

Data were analyzed using GraphPad Prism 5 (Graphpad Software, San Diego, CA, USA). As the data were in a non-normal distribution (base on Q-Q plots), the data were best represented by the median values. Data were analyzed with a Wilcoxon signed rank test with a two-tailed *p*-value (two groups) or a Friedman test with Dunn’s post-test when appropriate (three groups). The significance level α was 0.05.

## 3. Results

### 3.1. Patients

In the study, 14 treatment-naive RA patients (disease duration from diagnosis was 0 months) starting with MTX, and 12 RA patients starting with anti-TNF therapy in addition to MTX, were included (median RA disease duration was 19 months, range 6–370 months). In the MTX group, four patients were seronegative for both anti-CCP2 and IgM-RF, and two were seronegative for anti-CCP2. In the anti-TNF group, one patient was seronegative for both anti-CCP2 and IgM-RF, and one for only IgM-RF. Further patient characteristics, including intra-oral characteristics at baseline, are listed in Table 1. The median follow-up for the MTX group was 61 days (interquartile range (IQR 57–64)), and for the anti-TNF group it was 80 days (IQR 76–90) and 174 days (IQR 172–191) for the first and second follow-up visit, respectively. Two patients stopped with etanercept after the first follow-up visit because of the lack of efficacy on reducing RA activity, and continued treatment with tocilizumab or adalimumab. After the first follow-up visit, they were excluded from further analysis. One other patient in the anti-TNF group had to be excluded after the first follow-up visit because of the use of systemic antibiotics. 

### 3.2. Clinical Parameters

Treatment with MTX or MTX combined with anti-TNF both significantly improved DAS28 scores (Figure 1A). At baseline, the median DAS28 for the MTX group was 5.8 (IQR 3.8–6.5) at baseline, and after 2 months follow-up it was 3.2 (IQR 2.2–4.6) (*p* < 0.01). For the anti-TNF/MTX group, the median DAS28 was 5.3 (IQR 4.2–5.8) at baseline, after 3 months follow-up it was 2.8 (IQR 2.6–4.4), and after 6 months follow-up is was 3.3 (IQR 2.1–4.3) (*p* < 0.05).

The PISA scores did not change significantly following treatment with MTX (at baseline: median 1.7 cm^2^ (IQR 1.4–4), at T1: median 1.8 cm^2^ (IQR 1.4–3.1)) (Figure 1A). There was a decrease in the anti-TNF/MTX group regarding the PISA scores, although the decrease reached no significance (at baseline: median 1.9 cm^2^ (IQR 1.2–3.2), at T1: median 1.9 cm^2^ (IQR 1.0–3.3), and at T2: median 1.2 cm^2^ (IQR 0.8–2.2)). The level of oral hygiene did not change upon treatment (full-mouth plaque score at baseline: median 34% (IQR 24–65) for the MTX group, and median 36% (IQR 23–57) for the anti-TNF/MTX group, data not shown). No differences in clinical parameters were seen between smokers (*n* = 5) and non-smokers (*n* = 21), but a low number of smokers were studied (Appendix A). 

### 3.3. Laboratory Parameters

Treatment with MTX or anti-TNF/MTX significantly decreased the CRP and ESR levels (Figure 1B). At baseline, the median CRP and ESR levels for the MTX group were 19 mg/L (IQR 1.3–47) and 37 mm/h (IQR 12–77), respectively. After 2 months follow-up, these values decreased to 3.4 mg/L (IQR 0.3–10) and 11 mm/h (IQR 7.8–23), respectively (both *p* < 0.05). For the anti-TNF/MTX group, the median ESR level at baseline was 22 mm/h (IQR 10–56), after 3 months follow-up it was 13 mm/h (IQR 5.8–22), and after 6 months follow-up it was 7.0 mm/h (IQR 3.5–13, *p* < 0.05). The CRP levels in the anti-TNF/MTX group tended to decrease, but no significance was reached (median CRP level at baseline: 5.4 mg/L (IQR 2.9–21), after 3 months: 2.6 mg/L (IQR 0.7–9.6), and after 6 months: 2.9 mg/L (IQR 1.0–5.4)). The reduction in CRP and ESR levels was independent from the PISA or presence of other intra-oral pathologies, such as caries and peri-apical pathology. There were no significant changes in the anti-CCP2 and IgM-RF levels (anti-CCP2: at baseline the median was 122 U/mL (IQR 5–624) for the MTX group and 469 U/mL (IQR 191–901) for the anti-TNF/MTX group; IgM-RF: at baseline the median was 20 IU/mL (IQR 5–64) for the MTX group and 114 IU/mL (IQR 20–241) for the anti-TNF/MTX group; data not shown). The anti-*Pg* levels remained stable after treatment with MTX (at baseline: the median was 19 mg/L (IQR 1338)), but it increased after 6 months of anti-TNF treatment (at baseline: the median was 14 mg/L (IQR 4.328), and at T2: it was 18 mg/L (IQR 5.350, *p* < 0.01)) as shown in Figure 1C. The anti-*Pg* levels were not associated with any other laboratory parameter. 

## 4. Discussion

This longitudinal observational study assessed the influence of anti-rheumatic treatment (MTX, MTX plus anti-TNF) on the periodontal inflammatory burden, as measured by the PISA in RA patients. The results showed that anti-rheumatic treatment had negligible effects on the PISA, and thus, on periodontal inflammation. 

Regarding csDMARDs, our results, using the most sensitive current method to assess the systemic burden of periodontal disease, are in line with two other studies on this topic; both studies reported no effect in clinical periodontal parameters after starting csDMARD therapy (MTX, hydroxychloroquine, leflunomide, and sulfasalazine) in treatment-naive RA patients [22,23]. Ortiz et al. [22] assessed 10 RA patients after 6 weeks, and Äyräväinen et al. [23] assessed 46 RA patients after a median of 16 months; however, the follow-up period was not further specified. Thus, on basis of their and our data, it is unlikely that treatment with csDMARDs will have an effect on the periodontal condition. In the case that csDMARDS do have an effect on the periodontal condition when assessed in large patient cohorts, it is very unlikely that this effect will be clinically relevant. 

Regarding the influence of biological DMARD/anti-TNF therapy on clinical periodontal parameters, controversial results have been reported. Ziebolz et al. [14] reported that a combination treatment with MTX and anti-TNF resulted in an increased potential for periodontal inflammation, while we observed a negligible effect of this combination treatment on the clinical periodontal condition. Our observation that anti-TNF therapy had negligible effect on the clinical periodontal condition is in line with the observation of Ortiz et al. [22], who reported that anti-TNF therapy (infliximab, etanercept and adalimumab) without periodontal treatment had no significant effect on the periodontal condition. However, a comparable study in 18 Brazilian RA patients also found that periodontal parameters before and after starting anti-TNF therapy (infliximab, etanercept and adalimumab) remained stable throughout the study period of 6 months, while an improvement in the most analyzed RA parameters was observed [24]. In addition, Äyräväinen et al. [23] reported no effect on periodontal parameters in 26 RA patients starting treatment with biological DMARDs (mainly anti-TNF therapy; adalimumab, etanercept, golimumab and certolizumab pegol) after a median of 16 months. On the other hand, another report showed some beneficial effects: a Japanese study on 20 RA patients [25] showed some effect on bleeding scores and pocket probing depth after 3 months of anti-TNF therapy (adalimumab), although it is highly questionable whether the observed difference of 0.09 mm in the pocket probing depth is clinically relevant. Such a change in pocket probing depth is also well within the measuring error of repeated pocket probing depths. However, an effect of anti-TNF treatment on the periodontium of RA patients has been shown at the local level; anti-TNF can modify the host response in the inflammatory exudate of the periodontium [26,27]. This local effect may be too small to be reflected in clinically measurable and/or relevant changes, even when using the currently most sensitive method for assessing the inflammatory burden of periodontal disease, but it illustrates that RA patients taking immunosuppressive medications may be at increased risk to develop infections, e.g., periodontitis [10]. Although only two patients had cultivable *Pg* at baseline, it is generally accepted that many persons experience an infection with *Pg* during periodontitis and gingivitis. Moreover, it has been shown that serum levels of antibodies to *Pg* are extremely stable for many years, both in subjects with and without periodontitis, and that the antibody levels correlate with periodontitis and bacterium-positivity [28]. Our observation of increased serum levels of antibodies to anti-*Pg* after 6 months of anti-TNF therapy might be illustrative of this consequence. 

Anti-rheumatic treatment significantly decreased the disease activity of RA reflected by lower DAS28 scores, CRP levels, and ESR levels. This reduction in DAS28 was not related to the PISA or the presence of other intra-oral pathologies, such as caries and peri-apical pathology. On the contrary, periodontal treatment in RA patients was shown to clearly improve DAS28 scores and ESR levels, while other systemic inflammatory markers, such as CRP, IL-6, and TNF, tended to decrease [29,30].

*Pg* has been proposed to play a crucial role in the initiation or propagation of periodontitis-associated RA [21]. However, since the prevalence of subgingival *Pg* in our patient group was low, no relation of clinical or laboratory parameters to the current subgingival presence of *Pg* or other oral bacteria could be observed. Concordantly, Äyräväinen et al. [23] found that *Pg* was more often found in treatment-naive early RA patients. An effect of cs- and biological DMARDs on anti-CCP2 and RF levels was not observed. It has been shown that anti-CCP2 and RF levels may decrease after starting anti-TNF therapy, regardless of clinical response [31].

A limitation of our study was the low number of RA patients that were included in either treatment arm, although very well-defined from the perspective of expert rheumatologists and periodontologists. However, the number of patients needed to compare two groups was in line with the number of patients needed per group in the study of Ziebolz et al. [14], who used a cross-sectional design to compare seven treatments of RA with csDMARDS or biological DMARDS.

## 5. Conclusions

Within the limitations of this study, it can be concluded that anti-rheumatic treatment (MTX, and anti-TNF in addition to MTX) has a negligible influence on the periodontal condition of RA patients.

## Figures and Tables

**Figure 1 ijerph-18-02529-f001:**
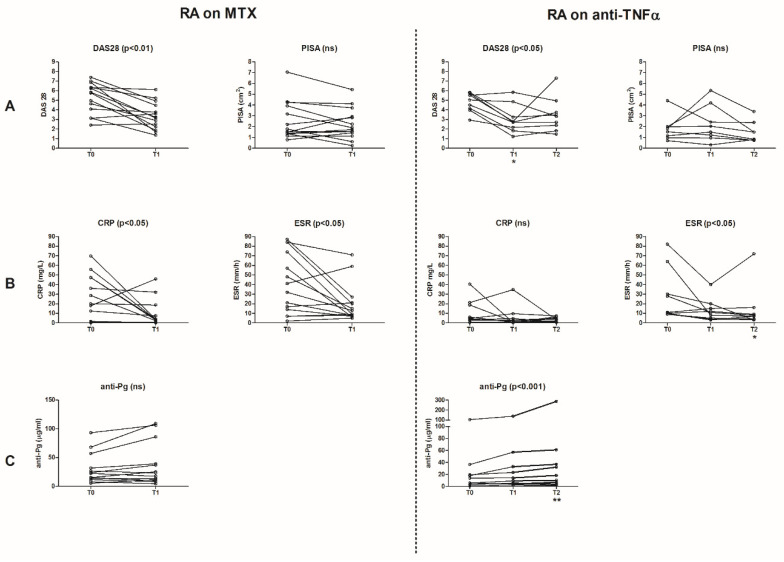
Effect of anti-rheumatic treatment (methotrexate (MTX) and anti-tumor necrosis factor-α (anti-TNF) in addition to MTX) on clinical and laboratory parameters of rheumatoid arthritis (RA) and periodontitis. Each line in the figures represents one patient over time in the study. (**A**) Clinical parameters: RA disease activity (DAS28) and periodontal inflamed surface area (PISA). (**B**) Laboratory parameters: erythrocyte sedimentation rate (ESR) and C-reactive protein (CRP) levels. (**C**) IgG antibody levels against *Porphyromonas gingivalis* (anti-*Pg*). T0: baseline, T1: after 2 months treatment of MTX and after 3 months treatment of anti-TNF in addition to MTX, T2: after 6 months treatment of anti-TNF in addition to MTX, ns: not significant.

**Table 1 ijerph-18-02529-t001:** Patient characteristics at baseline (IQR: interquartile range).

Patient Group	MTX	Anti-TNF
Number	14	12
Age in years (IQR)	61 (49–66)	64 (57–67)
Female gender (%)	71	75
Current smoker (%)	21	17
anti-CCP2 seropositive (>25 U/mL) (%)	57	92
IgM RF seropositive (>10 IU/mL) (%)	71	83
CRP (mg/L) median (IQR)	19 (1.3–47)	5.4 (2.9–21)
ESR (mm/h) median (IQR)	37(12–77)	22 (10–56)
DAS28 (IQR)	5.8 (3.8–6.5)	5.3(4.2–5.8)
**Intra-oral characteristics**		
Number of teeth (IQR)	26 (24–29)	25 (23–27)
Pocket probing depth ≥4 mm (% of sites per patient) (IQR)	8.0 (2.2–24)	8.5 (5–16)
Bleeding on probing (% of sites per patient) (IQR)	10 (7–23)	15 (11–21)
Periodontal inflamed surface area (PISA) in cm^2^ (IQR)	1.7 (1.4–4.0)	1.9 (1.2–3.2)
*Periodontal diagnosis (number of patients, %)*		
Gingivitis	8 (57)	8 (67)
Moderate periodontitis	4 (29)	4 (33)
Severe periodontitis	2 (14)	0
*Presence of other intra-oral pathologies (number of patients,%):*		
Furcation involvement	3 (21)	5 (42)
Peri-apical pathology	3 (21)	2 (17)
Caries	2 (14)	2 (17)
*Subgingival microbial characteristics (number of patients culture positive for,%):*		
*Porphyromonas gingivalis*	2 (14)	0
*Prevotella intermedia*	8 (57)	8 (67)
*Tannerella forsythia*	9 (64)	5 (42)
*Parvimonas micra*	13 (93)	10 (83)
*Campylobacter rectus*	8 (57)	4 (33)

## Data Availability

Data are available through the first author of the study.

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
