# Peer review of "Effect of Anti-Rheumatic Treatment on the Periodontal Condition of Rheumatoid Arthritis Patients"

_ijerph, 2021, doi:10.3390/ijerph18052529_

Round 1

Reviewer 1 Report

Major:

  • Introduction section regarding prevalence of periodontitis in patients with RA: The authors have stated the worldwide prevalence of RA in the introduction. However, what is not clear, is how common periodontitis is in this cohort. If possible, please include some numbers regarding the incidence and/or prevalence of periodontitis in patients with RA, as this may help to justify your small sample size. If not possible (e.g. due to lack of observational data), please state this in the author response.
  • Results/Discussion section regarding prevalence of Pg and anti-Pg levels: Only 2/26 patients grew Pg following baseline culture, but I understand that anti-Pg antibody levels were tested in the study design because Pg has been proposed to play a role in initiation of RA-associated periodontitis. This is a reasonable justification. However, what I am less clear on, is how the authors explain how levels of anti-Pg appear to increase (significantly so in the case of TNFi) in both treatment groups. It is particularly unclear what the scientific explanation behind this would be in the TNFi-treated group, as none of these patients cultured Pg at baseline. The authors need to include more discussion and explanation of this finding. Even if the answer is not known, it would improve the Discussion section of this manuscript even if the authors were to postulate a theory behind this, because at the moment, it feels as if this finding has been ignored in the Discussion after a very brief mention right at the end of the Results section.

Minor:

  • Lines 105-109 have been repeated from lines 78-82, and can be removed from the Methods section.
  • Section 2.3: Do you mean that peripheral blood was drawn at each follow-up visit, or was it drawn days or even weeks after each visit? If the latter is the case, are you able to explain why there was a delay, and the mean delay between follow-up visit and blood draw, as this may affect the accuracy of inter-patient calculation of DAS28?
  • Section 3.1: In the treatment-naïve RA patients, the authors state that disease duration is 0 months. Do you mean time from diagnosis? Or is this genuinely symptom duration and the Rheumatology department in which the study is based is extremely efficient at seeing new referrals? Please clarify this, either in your response to reviewers, or preferably, in the text of the manuscript.
  • Line 172: Given that you have mentioned differences in outcomes between smokers and non-smokers, it would be worth placing this data in Online Supplementary material, as some readers may find this interesting.
  • Figure 1: Please explain in the legend that each set of dots connected with lines represents a single patient’s outcomes over time, if indeed, this is what the figures actually represent. If not, please elaborate further in the legend.
  • Line 243: The authors mention periodontal treatment. What does this constitute of? As a non-periodontal specialist, I do not know what this consists of – I think it would be interesting to give a very brief description of what to expect here for non-specialists, or perhaps examples of treatment from the references that the authors have given.

Typographical errors:

  • Line 76: “Therefore, purpose of this study…” should read “Therefore, the purpose of this study…”
  • Line 218: For the abbreviation “MTA,” do you mean “MTX”?

Author Response

Introduction section regarding prevalence of periodontitis in patients with RA: The authors have stated the worldwide prevalence of RA in the introduction. However, what is not clear, is how common periodontitis is in this cohort. If possible, please include some numbers regarding the incidence and/or prevalence of periodontitis in patients with RA, as this may help to justify your small sample size. If not possible (e.g. due to lack of observational data), please state this in the author response.

Periodontitis is a rather common disease in patients with RA as about two thirds of the RA patients suffer from moderate or severe periodontitis (de Smit et al, 2012; Renvert et al., 2020). This prevalence number has been added to the introduction.

However, although periodontitis is rather common in RA patients, the number of subjects eligible for our study is much smaller as the patients had to be newly diagnosed (MTX), were >18 years, had to fulfill the ACR/EULAR criteria for RA (2010) and were willing to participate. Furthermore, the should have at least 6 teeth, and had no signs or history of  inflammation other than RA, present malignancy, diabetes, active thyroid disease, myocardial infarction, stroke or recanalisation of the femoral arteries for claudication shorter than 6 months prior to the study, pregnancy including a 6-months post-partum period, breastfeeding, malnutrition, alcoholism, drug abuse, use of corticosteroids >10mg/day, and antibiotic use shorter than 3 months prior to the study. In particular, antibiotic use <3 months prior to the study was common.

de Smit M, Westra J, Vissink A, Doornbos-van der Meer B, Brouwer E, van Winkelhoff AJ. Periodontitis in established rheumatoid arthritis patients: a cross-sectional clinical, microbiological and serological study.  Arthritis Res Ther. 2012 Oct 17;14(5):R222.

Renvert S, Berglund JS, Persson GR, Söderlin MK. The association between rheumatoid arthritis and periodontal disease in a population-based cross-sectional case-control study. BMC Rheumatol. 2020 Jul 20;4:31

Results/Discussion section regarding prevalence of Pg and anti-Pg levels: Only 2/26 patients grew Pg following baseline culture, but I understand that anti-Pg antibody levels were tested in the study design because Pg has been proposed to play a role in initiation of RA-associated periodontitis. This is a reasonable justification. However, what I am less clear on, is how the authors explain how levels of anti-Pg appear to increase (significantly so in the case of TNFi) in both treatment groups. It is particularly unclear what the scientific explanation behind this would be in the TNFi-treated group, as none of these patients cultured Pg at baseline. The authors need to include more discussion and explanation of this finding. Even if the answer is not known, it would improve the Discussion section of this manuscript even if the authors were to postulate a theory behind this, because at the moment, it feels as if this finding has been ignored in the Discussion after a very brief mention right at the end of the Results section.

Although only 2 patients had cultivable Pg at baseline it is generally accepted that many persons experience an infection with Pg during periodontitis and gingivitis. Moreover it has been shown that serum levels of antibodies to Pg are extremely stable during many years both in subjects with and without periodontitis and that the antibody levels correlate with periodontitis and bacterium-positivity.  (ref Lakio). The rise in levels of anti-Pg may indicate that a flare of periodontal disease may arise during treatment with TNF-inhibitors. We have added this information and the reference to the discussion.

Lakio L,  Antinheimo J, Paju S, Buhlin K, Pussinen P J, Alfthan G. Tracking of plasma antibodies against Aggregatibacter actinomycetemcomitans and Porphyromonas gingivalis during 15 years. J Oral Microbiol 2009, DOI: 10.3402/jom.v1i0.1979

Minor:

Lines 105-109 have been repeated from lines 78-82, and can be removed from the Methods section.

Thank you very much for noting this, these lines have been removed.

Section 2.3: Do you mean that peripheral blood was drawn at each follow-up visit, or was it drawn days or even weeks after each visit? If the latter is the case, are you able to explain why there was a delay, and the mean delay between follow-up visit and blood draw, as this may affect the accuracy of inter-patient calculation of DAS28?

The blood was drawn at each follow-up visit so there was no delay between blood measurements and calculation of DAS28.We have clarified this in the body of the text.

Section 3.1: In the treatment-naïve RA patients, the authors state that disease duration is 0 months. Do you mean time from diagnosis? Or is this genuinely symptom duration and the Rheumatology department in which the study is based is extremely efficient at seeing new referrals? Please clarify this, either in your response to reviewers, or preferably, in the text of the manuscript.

Indeed, we mean time of diagnosis. Patients were diagnosed and started as soon as possible with anti-rheumatic treatment, however, after the periodontal assessment at T0. Periodontal assessment was always performed shortly after diagnosis and the  anti-rheumatic treatment was started the same week the periodontal assessment was done. We have added to the body of the text that disease duration was counted from the time of diagnosis.

Line 172: Given that you have mentioned differences in outcomes between smokers and non-smokers, it would be worth placing this data in Online Supplementary material, as some readers may find this interesting.

We have added a supplementary figure showing the observed effects in smokers and non-smokers.

Figure 1: Please explain in the legend that each set of dots connected with lines represents a single patient’s outcomes over time, if indeed, this is what the figures actually represent. If not, please elaborate further in the legend.

Indeed each set of dots represents one patient over time. We have mentioned this in the legends.

Line 243: The authors mention periodontal treatment. What does this constitute of? As a non-periodontal specialist, I do not know what this consists of – I think it would be interesting to give a very brief description of what to expect here for non-specialists, or perhaps examples of treatment from the references that the authors have given.

The primary goal of periodontal treatment is to control microbial periodontal infection by removing the (subgingival) bacterial biofilm by mechanical debridement, and creating circumstances for optimal bacterial plaque control by the patient. Active periodontal treatment includes scaling and rootplaning with ultrasonic- or hand instruments, but can also include surgical therapy, extraction of hopeless teeth and/or use of local or systemic antiseptic or antibiotic measures. Periodontal therapy triggers a short-term inflammatory response followed by a progressive and consistent reduction of systemic inflammation (d‘Aiuto et al. 2013).

We have such an explanation to the materials and methods section.

D'Aiuto, F.; Orlandi, M.; Gunsolley, J. C.  Evidence that periodontal treatment improves biomarkers and CVD outcomes J. Clin. Periodontol. 2013,;40 Suppl 14, S85-S105.

Line 76: “Therefore, purpose of this study…” should read “Therefore, the purpose of this study…”

Line 218: For the abbreviation “MTA,” do you mean “MTX”?

Thanks for mentioning the typographical errors in lines 76 and 218, they have been corrected.

Reviewer 2 Report

The Authors must see my remarks

Author Response

Thanks for the notes you added to the manuscript. We revised the manuscript according to your suggestions. We added also our answers to your remarks (see pdf).

Reviewer 3 Report

Thanks for sending the manuscript over to me. The manuscript presents an important and unexplored area of health research.  However, I have some comments that can help to improve the quality of the manuscript-

  1. The abstract does not have enough methodological information. Add some more methodological information.
  2.  The reasons for doing this research are not clearly elaborated in the introduction part.  You can also include the following research paper in the introduction Periodontal connections to the coronavirus disease 2019: An unexplored novel path? Shaikh MS, Lone MA, Kabir R, Apu EH - Adv Hum Biol (aihbonline.com)
  3. The inclusion and exclusion criteria are not clearly stated. 
  4. What was the study duration? Which year the data was collected?
  5. How the clinical parameters information was recorded? How did they deal with the uncooperative patients? 
  6. Page 3 line 116, mentioned that a standard sampling procedure was used. What is that standard sampling procedure?
  7. Why median age was calculated here? 
  8. Why the baseline data is not compared with the follow-up data? The findings can be presented in a tabular format.
  9. The discussion lacks a balanced argument.

Author Response

Thanks for sending the manuscript over to me. The manuscript presents an important and unexplored area of health research.  However, I have some comments that can help to improve the quality of the manuscript.

Thank you very much for you comments. Your comments and the comments of the other reviewers have helped to improve the quality of or manuscript.

1.The abstract does not have enough methodological information. Add some more methodological information.

We have added the following sentence to the abstract: ”Also, erythrocyte sedimentation rates were determined and levels of C-reactive protein, IgM-rheumatoid factor, anti-cyclic citrullinated protein antibodies, and antibodies to the periodontal pathogen Porphyromonas gingivalis were measured. Subgingival sampling and microbiological characterization of the subgingival microflora was done at baseline.”

  1. The reasons for doing this research are not clearly elaborated in the introduction part. You can also include the following research paper in the introduction Periodontal connections to the coronavirus disease 2019: An unexplored novel path? Shaikh MS, Lone MA, Kabir R, Apu EH - Adv Hum Biol (aihbonline.com)

The purpose of the study was to investigate whether antirheumatic treatment had any effect, positive of negative, on the periodontal condition of RA patients. Also, the influence of the periodontal condition on the effectiveness of anti-rheumatic treatment could by investigated. Rheumatologists should be informed if the periodontal condition of a patient would be worsened by a specific treatment or if periodontal treatment had to be done. As asked for, we added a statement good oral health is important for a patient’s wellbeing, and thus information on the effects of treatment is important.

With regard to the paper supposed, this was an interesting letter to the editor to read, but we do not feel that this paper adds to the message in our paper other than that this paper stresses that the periodontium can play an important role in general health and GCF can be used as a non-invasive diagnostic tool.

3.The inclusion and exclusion criteria are not clearly stated.

The following lines are mentioned under section 2.1 ‘Patients’.

Inclusion criteria were >18 years, fulfilling the ACR/EULAR criteria for RA (2010) [16] and willingness to participate and written informed consent. Exclusion criteria were edentulism, infection other than periodontitis, inflammation other than RA, present malignancy, diabetes, active thyroid disease, myocardial infarction, stroke or recanalisation of the femoral arteries for claudication shorter than 6 months prior to the study, pregnancy including a 6-months post-partum period, breastfeeding, malnutrition, alcoholism, drug abuse, use of corticosteroids >10mg/day, antibiotic use shorter than 3 months prior to the study.

4.What was the study duration? Which year the data was collected?

The study duration was approximately one year (2016).

5.How the clinical parameters information was recorded? How did they deal with the uncooperative patients?

The rheumatological clinical parameters were recorded by a rheumatology research nurse. Periodontal clinical parameters were recorded by a periodontist. This is described in section 2.2. ‘Clinical parameters’. One of the inclusion criteria was that patients had to be willing to participate. This is described in section 2.1 ‘Patients’.

6.Page 3 line 116, mentioned that a standard sampling procedure was used. What is that standard sampling procedure?

Subgingival samples were taken by using sterile paper points. Microbiological sampling included selection of the deepest bleeding periodontal pocket in each quadrant of the dentition on the basis of pocket probing depth measurements. If there were no bleeding periodontal pockets, mesial sites of the first molars or, in absence of a first molar, the mesial site from the adjacent anterior tooth in the dental arch was selected. Sample sites were isolated with cotton rolls, and supragingival plaque was carefully removed with curettes and cotton pallets. Subsequently, two paper points were inserted to the depth of the pocket and left in place for 10 seconds. All paper points per subject were pooled in reduced transport fluid and processed for microbiological examination immediately after sampling.

Smit, M.; Westra, J.; Vissink, A.; Doornbos-van der Meer, B.; Brouwer, E.; van Winkelhoff, A. J. Periodontitis in established rheumatoid arthritis patients: a cross-sectional clinical, microbiological and serological study. Arthritis Res. Ther. 2012, 14, R222.

7.Why median age was calculated here?

We showed median and interquartile range when data were not normally distributed. Age of our patient cohorts were not normally distributed.

8.Why the baseline data is not compared with the follow-up data? The findings can be presented in a tabular format.

The data that were investigated at T0, T1 and T2 are shown and compared in the figures and are mentioned in the text under results.

Round 2

Reviewer 3 Report

The authors have done extensive changes to their manuscript.